# Comparative readability of information on different treatment options for breast cancer, based on WeChat public accounts

**Bingyan Li[1], Jia Liu[2], Yuxi Zhang[2], Wenjuan Yang[2], Min Liu[1,2], Lunfang Xie[2]***

**1** The First Affiliated Hospital of Anhui Medical University, Hefei, China, **2** School of Nursing, Anhui Medical University, Hefei, Anhui, China

* 527548725@qq.com

## Abstract

### Objective

To evaluate and compare the readability of information on different treatment options for breast cancer from WeChat public accounts, propose targeted improvement strategies based on the evaluation of the results of the various treatment options, and provide a reference for producers of WeChat public accounts from which to write highly readable information regarding breast cancer treatment options.

### Methods

With "breast cancer" as keywords in April 2021, searches were implemented on Sogou WeChat website (https://weixin.sogou.com/) and WeChat mobile app. The selected WPAs were aimed to provided breast cancer health information, and the four latest articles of each WPA were included in the evaluation. Two independent observers assessed the readability of the articles through the Suitability Assessment of Materials (SAM) tool, and compared the readability of information on different treatment options, i.e., surgical treatment, medical treatment, complementary and alternative medicine (CAM), and comprehensive treatment.

### Results

A total of 136 articles on different types of breast cancer treatments from 37 WeChat public accounts were included in the present study. The median SAM score was 50 (IQR, 41–60). In terms of treatment options, the readability of articles in the CAM category scored higher in the content 75 (IQR, 63–81), learning stimulation and motivation 75 (IQR, 50–83) and cultural appropriateness 75 (IQR, 75–75) categories than in the medical and surgical treatment categories (P < 0.05). Additionally, the readability of articles in the CAM category scored higher in the cultural appropriateness 75 (IQR, 75–75) category than those for comprehensive and medical treatment ($P < 0.05$).

**Data Availability Statement:** All relevant data are within the manuscript and its Supporting Information files.

**Funding:** This work was supported by the Key project of natural science research in Colleges and universities of Anhui Province (grant number KJ2021A0255) awarded to LX; the Anhui Medical University School of Nursing Graduate Youth Program Cultivation Project (grant number hlqm12023038) awarded to BL; and the Key Laboratory of Geriatric Long-term Care (Naval Medical University) Ministry of Education (grant number LNYBPY-2023-05) awarded to LX.

**Competing interests:** There are no patents, products in development or marketed products associated with this research to declare.

## Conclusions

The overall readability of information on breast cancer treatment options in WeChat public accounts was in the lower portion of the "adequate" level. The readability of articles on medical treatment options is poor, especially on clinical trial articles, which could be improved in terms of content, graphics, learning stimulation, and motivation to make them more suitable for public reading.

## Introduction

According to Global Cancer Observatory (GLOBOCAN) 2020 data, breast cancer in women has surpassed lung cancer as the most common cancer in 2020, with 2,261,419 new cases, accounting for 11.7% of all cancers [1]. In China, breast cancer is an important cause of disability and death in women. According to 2020 data, the number of patients diagnosed with breast cancer in China was approximately 416,000, accounting for 9.1% of cancer cases nationwide, and having the fourth highest mortality rate among Chinese women [2]. In recent years, the quality of healthcare in China has been improving. However, the overall incidence and mortality rate of breast cancer are still increasing, leading to a focus on the prevention and control of malignant tumors in China.

For patients with breast cancer, quality care should include supportive measures, such as recommending quality sources of online health education information, to help them cope with their diagnosis [3]. The results of a study of patients with breast cancer in Italy showed that there is a high demand for online health education information, specifically in regards to treatment options [4]. The primary treatment options for breast cancer include surgery, medical treatment, complementary and alternative medicine (CAM), and comprehensive treatment. Medical treatment includes chemotherapy, radiotherapy, endocrine therapy, targeted therapy, and various clinical trials (chemotherapy and endocrine therapy) [5]. CAM refers to a group of medical practices and products that are not considered to be conventional medicine or standard treatments, with "complementary therapies" used in conjunction with conventional treatments and "alternative therapies" to replace them [6]. Comprehensive treatment involves the use of two or more surgical treatments, medical treatment, and/or CAM, i.e., chemotherapy and surgery. Most patients with breast cancer, however, lack the necessary knowledge regarding disease treatment options and the side effects associated with these options to make proper treatment decisions [7].

With the development of mobile networks, social media has become an important source of health information for breast cancer patients [8]. WeChat public accounts (WPAs) of WeChat have become an indispensable information dissemination platform for Chinese government agencies, medical institutions, enterprises and individuals [9–11], and gained 360 million readers and over 20 million registered accounts [10]. In Western countries, 29%–57% of breast cancer patients have searched the Internet for health-related information [12–14]. Zhang X surveyed 1,636 Chinese WeChat users and found that 97.68% had read health information via WeChat, and 53.36% mainly obtained health information via WPAs [9]. Furthermore, it was reported that the patients with breast cancer paid long-term attention to the related WPAs [15–19].

Readability is an important indicator for assessing whether the material can be effectively read and understood by readers [20]. Health messages with high readability are beneficial to help patients remember medical information [21], improve treatment compliance [22], and

facilitate doctor-patient communication [23]. Conversely, materials with low readability can cause difficulty for patients in understanding treatment options and increase their negative emotions [24]. For individuals with breast cancer, factors related to the layout and design of the reported health information may support the comprehensibility of patient education materials (PEMs) [25]. However, people are increasingly skeptical about the readability of health information released through WPAs [26]. To ensure the authenticity and security of WPAs, Tencent provides authentication services for WPAs. The authentication information and the WeChat Authentication Unique Identity are displayed in the authentication details of the account. However, authentication is not mandatory.

Currently, the readability of health information on topics such as mastectomy, breast cancer risk assessment tools, and postoperative breast reconstruction has been evaluated. However, their overall readability levels were beyond the officially recommended sixth grade reading level, which makes it difficult for the public to read and understand [7,15,27–32]. However, there has been no previous research undertaken to assess and compare the readability levels of different types of breast cancer treatment information in WPAs. The present study, therefore, assessed the readability of information available in WPAs regarding different treatment options in terms of non-textual factors for breast cancer to clarify the dimensions of relatively high and low readability of each type of information, and to help WPA producers focus on improving the readability of this information. This article is part of our study named "quality and readability evaluation on breast cancer treatment information on WPAs" and another paper about the quality of information has been published [33].

## Methods

### Sample selection

On April 11, 2021, the observer (WY) first retrieved articles and WPAs with the keywords "breast cancer" from both Sogou WeChat website (https://weixin.sogou.com/) and WeChat client and obtained WPAs by tracing the publisher of articles. We included the WPAs those WeChat profile pages clearly indicated that they provided health information for patients with BC. The exclusion criteria were as follows: the WPAs (1) were duplicate; (2) had not updated for more than one year; (3) delivered only academic research information to professionals; (4) whose target population were not only patients with BC. Two independent observers (BL and WY) screened the retrieved WPAs according to the information in WPA's profile page. Any discrepancies between the observers were resolved by discussing with a third author (ML) for consensus. Of 1332 retrieved WPAs, 41 were included in the analysis.

The article sample pool was formed by taking four articles that were newly released by each WPA on the survey date (April 27, 2021) for a total of 164 articles. Next, two

observers (BL and WY) independently selected the articles and any disagreements were resolved with an open discussion and consensus. Articles which provided information on breast cancer treatment options were included in the present study, while the exclusion criteria were as follows: (1) duplication; (2) only providing pictures, videos, and links to articles; (3) content of academic articles not read by the public; and (4) non-chinese articles. Finally, 136 articles published by 37 WPAs were included. Four WPAs were further excluded because they lacked breast cancer treatment articles. Fig 1 illustrates the search process. No patient health information was involved in the 136 articles included in this study.

### Evaluation tools

**Basic characteristics information table.**   The following information was collected: WPA name, article title, and treatment options. WPA owners, as listed on the WPAs' profile page,

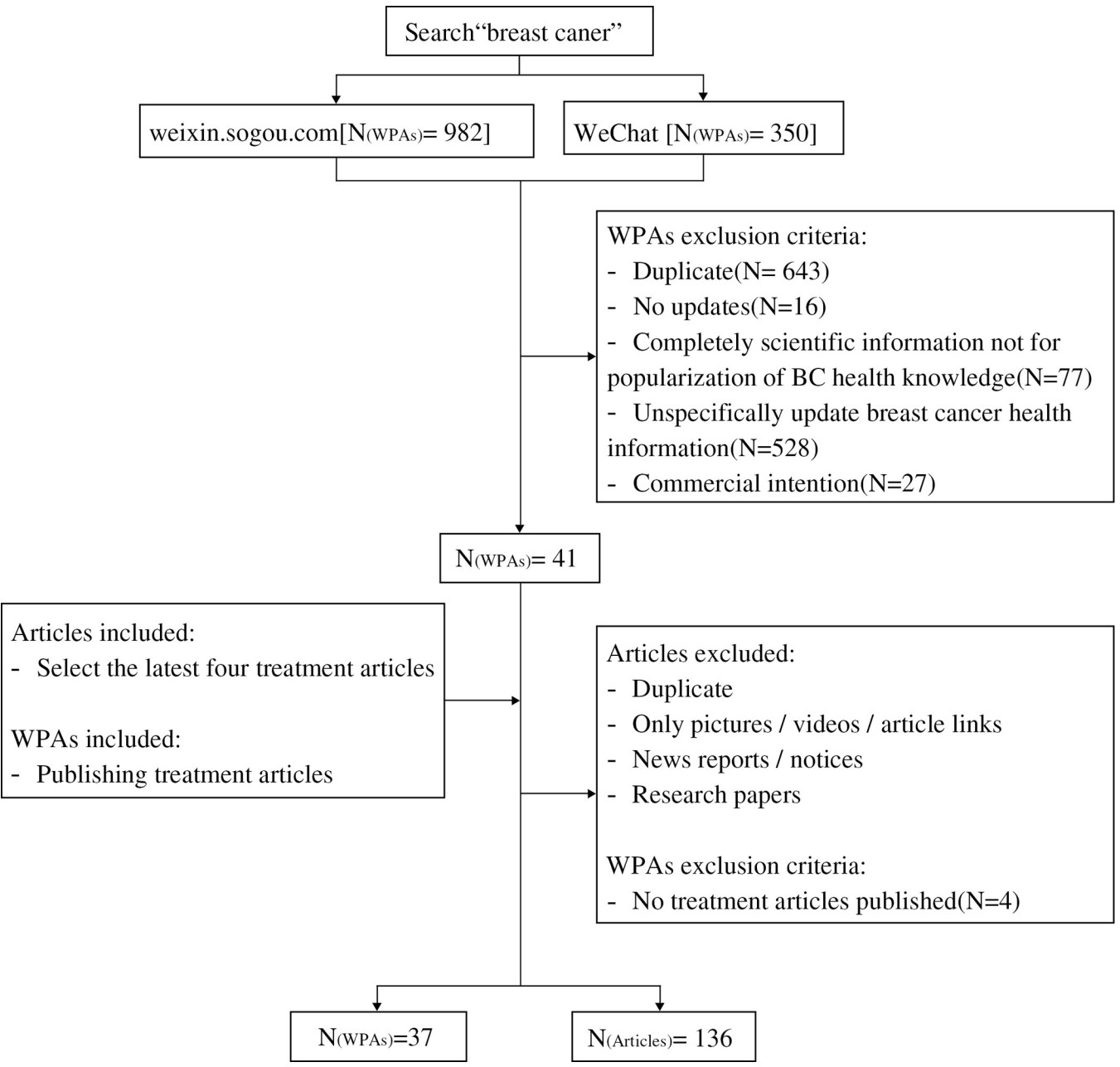

**Fig 1. Screening process of WeChat public accounts (WPAs) and breast cancer treatment-related articles.**

were categorized into individuals (e.g., physicians), enterprises (e.g., companies), institutions (e.g., public hospitals), and non-profit organizations (e.g., academic conferences). Breast cancer treatment options are classified as surgery, medical treatment (eg, radiation therapy, chemotherapy, endocrine therapy, targeted therapy, clinical trials, other), CAM (eg, complementary therapies, alternative therapies), and comprehensive treatment. Further, we identified several subgroup if a treatment option was classified as medical treatment or CAM. For example, if the treatment option was classified as a "CAM", we would further code it as a "complementary therapy" or "alternative therapy". The detailed information about 37 WPAs and 136 articles were provided in the S1 File.

**The Suitability Assessment of Materials (SAM).** The Suitability Assessment of Materials (SAM) was created by Doak et al in 1996 [34]. This tool is widely used to assess the readability of online health education materials [35–38], and is based on a scale, totaled from 22 items, consisting of the following 6 dimensions: content (4 items), literacy demand (5 items), graphics (5 items), layout and typography (3 items), learning stimulation and motivation (3 items), and cultural appropriateness (2 items). Each item is rated from 0 to 2 (0, "not suitable"; 1, "adequate"; 2, "superior"), and if there is something that cannot be assessed, based on the criteria of that item, it is rated as not suitable (N/A), and 2 points are subtracted from the total score. In 2013, the tool was translated into Chinese by Xianwen Li [39]. In the original scale, the Flesch-Kincaid readability formula was used to assess the "reading level" of the literacy demand dimension. As there was no readability formula in China to match this item, the item was deleted in the Chinese version of SAM. So we used the tool consisted of 21 items to evaluate the readability of the 136 articles included in the present study. The dimension and total scores of the scale were calculated as a score rate (material score/total score × 100%) and were divided into the following three levels: superior (70–100%), adequate (40–69%), and not suitable (0–39%). See S2 File for dimension and item details of the Chinese version of the Suitability Assessment of Materials (SAM). The overall inter-rater reliability of the SAM scale in the present study was 0.75, indicating substantial agreement [40], while the dimensions ranged from 0.69–0.9.

## Rating process

Two postgraduate medical students (ML and WY) with extensive assessment experience assessed 136 articles referencing the SAM Handbook. Five identical articles were firstly evaluated by the two observers in order to ensure their inter-rater consistency. To maintain the integrity of the data, the researchers sorted the 136 included articles by serial number and then shared them sequentially in a WeChat group. To simulate a real-life reading situation, the evaluations were conducted on the WeChat app. The two observers used SAM to evaluate the readability of all the articles. If there were inconsistencies in the score of an item between the two observers, they attended to reach a consensus through discussion. See S3 File for item and dimension scores of 136 articles based on different treatment options and subgroups.

## Statistical analysis

SPSS 27.0 for Windows (IBM, Chicago, IL, USA) and Excel 2016 software were used to analyze the data in the present study. The Shapiro-Wilk test was used to determine whether the data were normally distributed. The measurement data in the present study are expressed as median and interquartile range (IQR) due to non-normal distribution, while the count data are described as frequency, composition ratio, and rate. Readability scores for various breast cancer treatment regimens were compared using the rank sum test, which were also used to compare different medical treatment subgroups and CAM protocols separately, and multiple comparisons were performed using the Kruskal-Wallis test (K-W test). Differences were considered statistically significant at $P < 0.05$.

## Results

### Characteristics of the included WPAs and articles

Among the 37 WPAs, individual users accounted for the largest share (20, 54%), businesses (8, 22%), institutions (6, 16%), and nonprofit organizations (3, 8%). Of the 136 articles included, 74 (54%) were in regards to medical treatments, accounting for the highest percentage, with 27

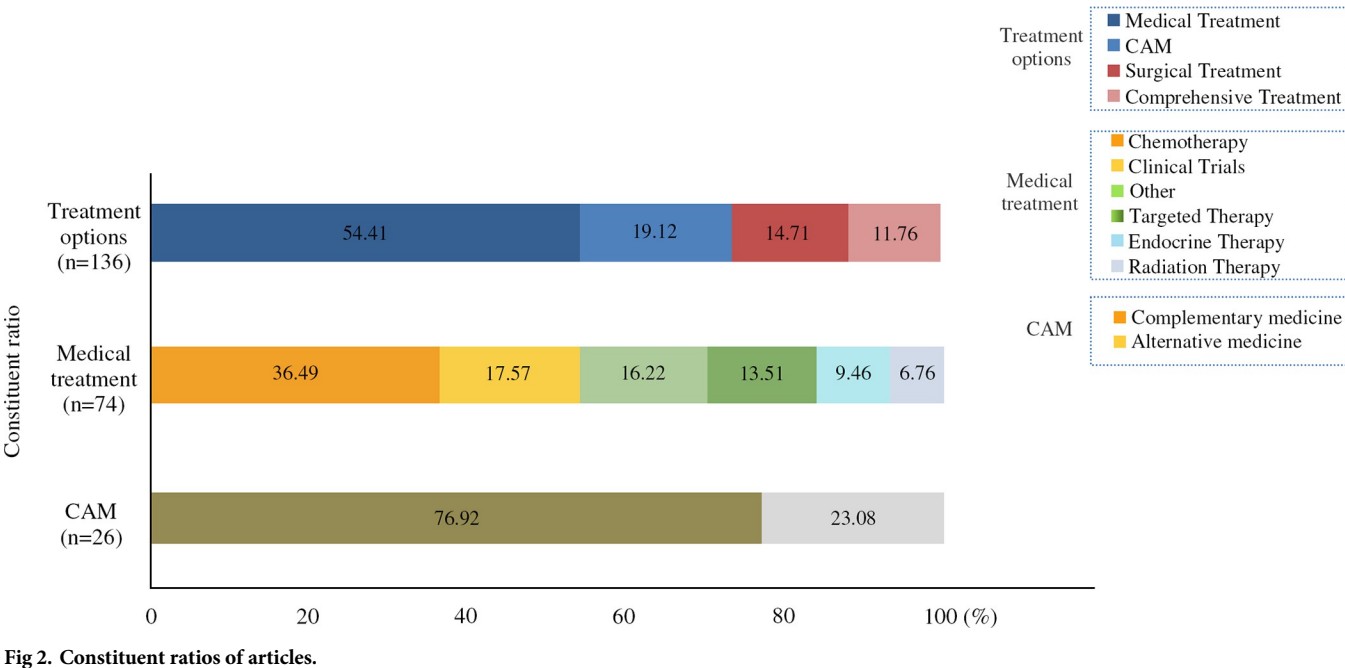

**Fig 2. Constituent ratios of articles.**

(36%) on chemotherapy and 26 (19%) on CAM treatments, of which 76% were complementary. Fig 2 shows the details the composition of the articles.

## Overall readability score

The SAM scores of 136 articles included in the present study ranged from 14 to 76, with a median score of 50 (IQR, 41–60), indicating that the overall readability of health information on breast cancer treatment was "adequate". The 3 of the articles (2%) were rated as "superior", 104 (77%) were rated as "adequate", and 29 (21%) were rated as "not suitable". According to the scores for each dimension, the readability of layout and typography was rated as "superior", while the content, literacy demand, learning stimulation and motivation, and cultural appropriateness were rated as "not suitable".

## Comparison of the readability of articles regarding different treatment options

In regards to the total score, the readability of the CAM articles was higher than that of the medical ($P < 0.001$) and comprehensive ($P = 0.03$) treatment articles. A comparison of the total readability scores and the scores for each dimension of the articles presenting different treatment options is seen in Table 1.

In terms of dimensional scores, the differences among the scores for content, learning stimulation and motivation, and cultural adaptability were statistically significant ($P < 0.05$), with the readability rating of the medical treatment articles being "not suitable" for learning stimulation and motivation. Fig 3 shows the details of the SAM scores for the different types of breast cancer treatments. For the content dimension, the readability scores of the CAM articles were higher than those of the medical ($P < 0.001$) and surgical ($P = 0.04$) treatment articles. In the learning stimulation and motivation dimension, the readability scores of the CAM articles were higher than those of the medical ($P = 0.001$) and surgical ($P = 0.04$) treatment articles.

**Table 1. Comparison of SAM scores across dimensions based on treatment options.**

| Category | Median SAM Score (IQR) | | | | | | |
|---|---|---|---|---|---|---|---|
| | Sum scores | Content | Literacy demand | Graphics | Layout and typography | Learning stimulation and motivation | Cultural appropriateness |
| Surgical treatment | 51 (48–55) | 50 (38–75)* | 50 (38–63) | 40.(30–50) | 83 (67–83) | 50 (21–63)* | 63 (50–75) |
| Medical treatment | 48 (38–57)* | 50 (38–63)* | 50 (34–63) | 40 (20–50) | 83 (67–83) | 33 (33–50)* | 50 (50–75)* |
| CAM | 63 (51–67) | 75 (63–81) | 56 (38–63) | 40 (38–50) | 83 (79–83) | 75 (50–83) | 75 (75–75) |
| Comprehensive treatment | 46 (39–60)* | 50 (41–72) | 50 (28–63) | 40 (23–40) | 83 (83–83) | 41 (33–67) | 50 (39–60)* |
| $H$ | 18.217 | 22.777 | 2.787 | 2.311 | 1.317 | 15.375 | 15.922 |
| $P^a$ | < 0.001 | < 0.001 | 0.426 | 0.510 | 0.725 | 0.002 | 0.001 |

*: Compared with CAM treatment, $P < 0.05$.

a: P-values were calculated with the Kruskal-Wallis H test.

And in the cultural appropriateness dimension, the readability scores of the CAM articles were higher than those of the medical ($P = 0.01$) and comprehensive ($P = 0.013$) treatment articles.

Table 2 shows a comparison of the median SAM scores of each dimension for the articles discussing medical treatment subgroups. In terms of overall scores, the readability scores of articles regarding clinical trials were lower than those of chemotherapy ($P < 0.001$) and radiotherapy ($P = 0.03$) articles. In terms of dimensional scores, the differences among the scores for content, literacy demand, graphics, learning stimulation and motivation, and cultural appropriateness were statistically significant ($P < 0.05$), with a readability rating of "not suitable" for clinical trial therapy articles for the content, graphics, learning stimulation, and motivation dimensions. The readability rating of the targeted therapy articles was "not suitable" for the literacy demand dimension.

Fig 4 shows the SAM scores for each dimension for the articles regarding the different medical treatment subgroups. For the content dimension, the readability score of clinical trial articles was lower than that of chemotherapy articles ($P = 0.006$). For the literacy demand dimension, the readability score of chemotherapy articles was higher than that of targeted

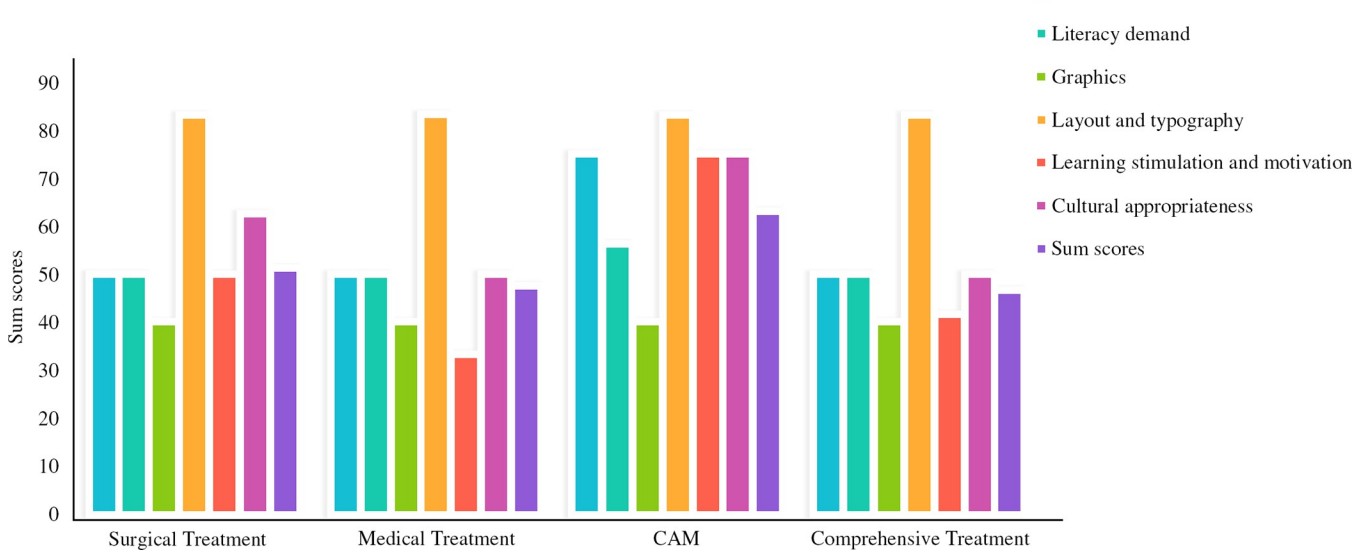

**Fig 3. Grades for SAM scores based on treatment options (n = 136).**

**Table 2. Comparison of SAM scores across dimensions based on medical treatment subgroup.**

| Category | Median SAM Score (IQR) | | | | | | |
|---|---|---|---|---|---|---|---|
| | Sum scores | Content | Literacy demand | Graphics | Layout and typography | Learning stimulation and motivation | Cultural appropriateness |
| Radiation therapy | 60 (44–63)*** | 63 (50–75) | 63 (56–69) | 50 (15–50) | 83 (83–83) | 50 (50–67)*** | 75 (50–75) |
| Chemotherapy | 55 (48–64)*** | 50 (38–75) | 63 (50–63) | 50 (40–60) | 83 (67–83) | 50 (33–67)*** | 75 (50–75) |
| Endocrine therapy | 43 (31–55) | 38 (13–63) | 50 (25–50) | 40 (30–50) | 83 (83–83) | 50 (33–50) | 50 (25–75) |
| Targeted therapy | 45 (28–52) | 44 (25–63) | 38 (13–53)** | 40 (28–50) | 83 (75–83) | 33 (17–50) | 50 (25–56)** |
| Clinical trials | 38 (25–40) | 38 (25–50)** | 38 (25–56) | 20 (10–25)** | 67 (50–83) | 33 (17–33) | 50 (25–75) |
| Other | 45 (35–52) | 38 (25–69) | 50 (25–59) | 40 (23–50) | 83 (71–83) | 33 (33–50) | 50 (25–69) |
| $H$ | 24.705 | 13.866 | 16.914 | 15.036 | 8.669 | 17.994 | 13.912 |
| $P^b$ | < 0.001 | 0.016 | 0.005 | 0.010 | 0.123 | 0.003 | 0.016 |

\*\*: Compared with chemotherapy, $P < 0.05$.

\*\*\*: Compared with clinical trials, $P < 0.05$.

b: P-values were calculated with the Kruskal-Wallis H test.

therapy articles ($P = 0.005$). For the graphics dimension, the readability score of the clinical trial articles was lower than that of the chemotherapy articles ($P = 0.002$). For the learning stimulation and motivation dimension, the readability scores were lower for the clinical trials articles than for the chemotherapy ($P = 0.003$) and radiation therapy ($P = 0.048$) articles. And for the cultural appropriateness dimension, the readability score of the chemotherapy articles was higher than that of targeted therapy articles ($P = 0.018$).

Table 3 shows a comparison of the median SAM scores of each dimension for the articles discussing the different CAM subgroups. The readability scores of the complementary therapy articles were higher than those of the alternative therapy articles. Fig 5 shows the SAM scores of the different CAM subgroups. Complementary therapy articles had a higher readability

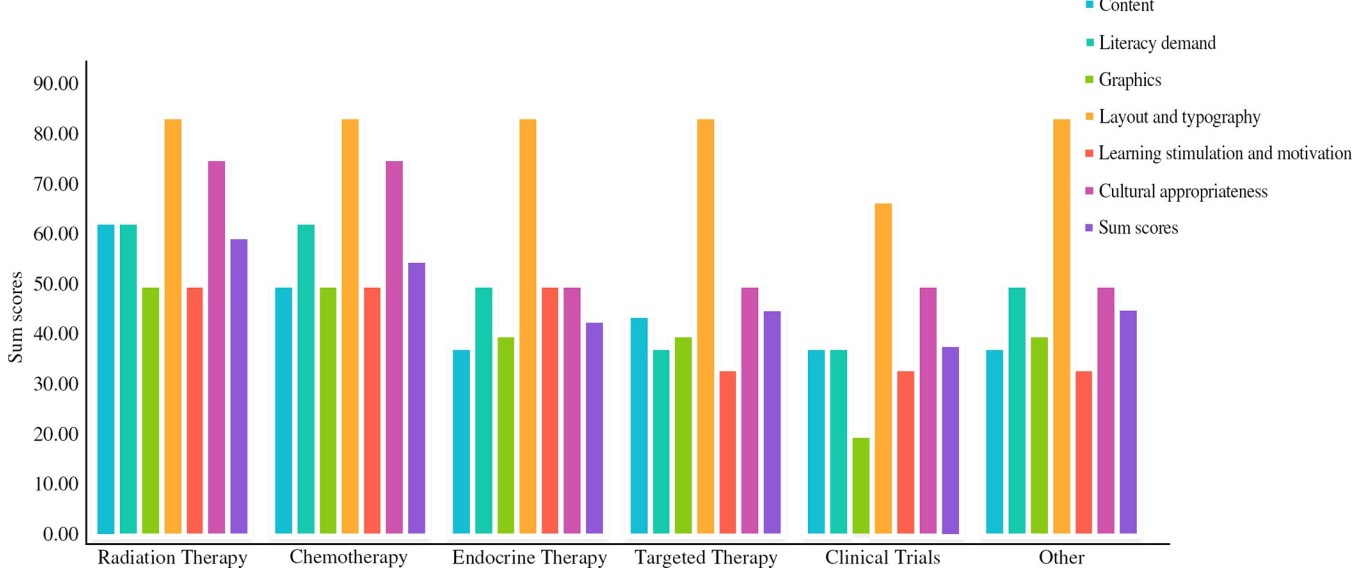

**Fig 4. Grades for SAM scores based on medical treatment subgroup (n = 74).**

**Table 3. Comparison of SAM scores across dimensions based on CAM subgroup.**

| Category | SAM score Median (IQR) | | | | | | |
|---|---|---|---|---|---|---|---|
| | Sum scores | Content | Literacy demand | Graphics | Layout and typography | Learning stimulation and motivation | Cultural appropriateness |
| Complementary medicine | 64 (60–68) | 75 (63–94) | 63 (50–72) | 40 (33–50) | 83 (71–83) | 83 (54–83) | 75 (75–75) |
| Alternative medicine | 46 (40–57) | 56 (38–81) | 25 (22–41) | 45 (38–55) | 83 (75–83) | 50 (42–63) | 50 (25–75) |
| Z | -2.294 | -1.710 | -3.403 | -0.408 | -0.289 | -1.518 | -3.021 |
| $P^c$ | 0.022 | 0.087 | 0.001 | 0.683 | 0.772 | 0.129 | 0.003 |

c: P-values were calculated with the Mann-Whitney U test.

score than the alternative therapy articles for the literacy demand ($P = 0.001$) and cultural appropriateness dimensions ($P = 0.003$), with alternative therapy articles rated as "not suitable" for the literacy demand dimension.

## Discussion

To our knowledge, this is the first study to assess and compare the readability of different treatment types of information for breast cancer in WPAs. It was found that the readability of information about breast cancer treatment options in WPAs was "adequate" and there were some significant differences in some dimensions between the articles on different types. The results of this study could be seen as a valuable reference for the authors and editors of treatment information in WPA to improve readability.

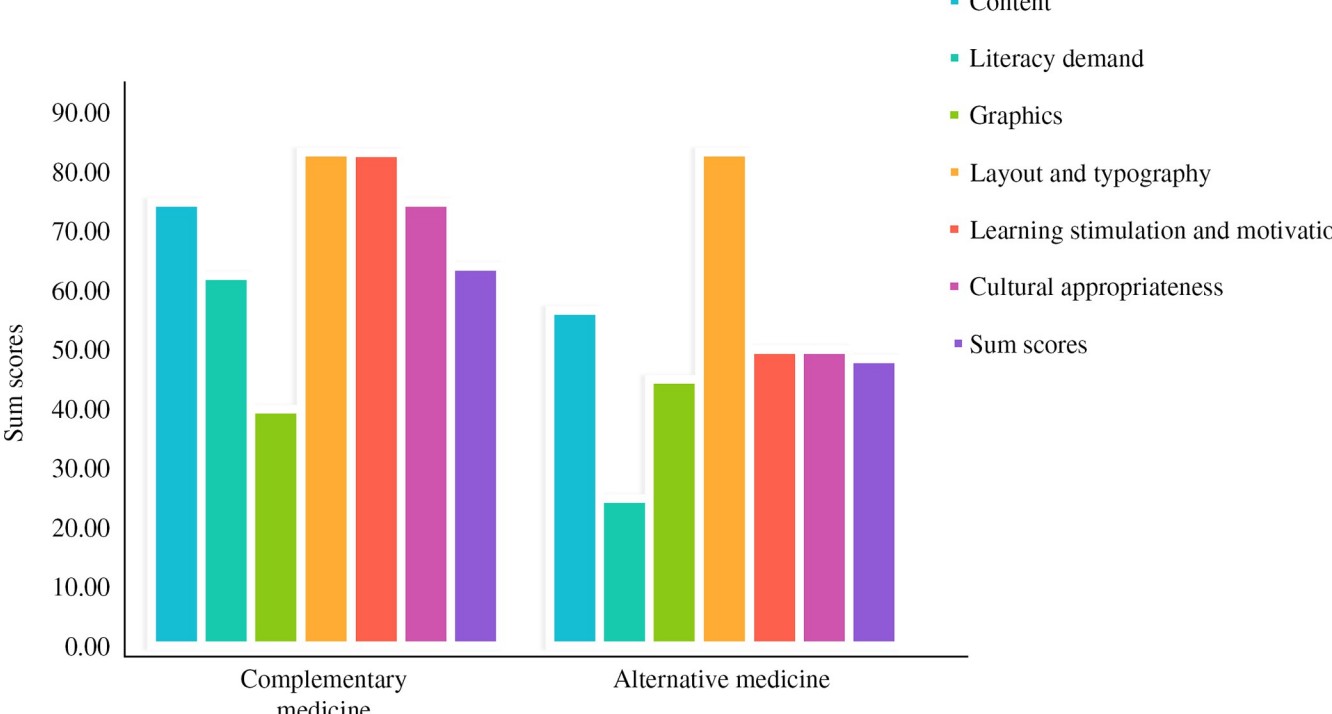

**Fig 5. Grades for SAM scores based on CAM subgroup (n = 26).**

Of the 136 articles included in the present study, 74 (54%) described medical treatment options, with chemotherapy accounting for 27 of those, the highest percentage (36%). Patients with breast cancer, in every stage except 0, require medical treatment at some point, especially chemotherapy [41,42]. According to a previous survey, one of the most common forms of systemic treatment for breast cancer is chemotherapy, with as many as 80% of breast cancer patients receiving chemotherapy [43,44]. The large number of breast cancer patients receiving chemotherapy creates a high demand for health information about this treatment, which may be the reason that chemotherapy encompasses the majority of medical treatment-related articles the present study. Complementary therapies were discussed in 20 of the CAM treatment articles, the highest percentage (77%). Fatmah studied the use of CAM among breast cancer patients in Saudi Arabia, and found that complementary therapies, such as diet and nutritional supplements, were the most widely used component of CAM, which is consistent with the results of the present study [45]. The high percentage of articles related to complementary therapies may be explained by the fact that dietary recommendations are easier for breast cancer patients to learn and put into practice than alternative therapies, and therefore reading interest is higher [46].

The SAM scale was used to assess readability, and 29 of the 136 articles (21%) were rated as "unsuitable". Analysis of the reasons found that most articles did not clearly state the purpose of the graphics to the reader, graphics were not closely related to the text, and the articles lacked captions for the graphics or did not explain the diagrams. A previous study showed that readers could remember an additional 35% of the information they read, given a combination of graphic and text health information [47,48]. Therefore, the suggestions were proposed as follows: the article cover should state the purpose of the articles to the readers, the article content should be illustrated with pictures which should be clear and captioned and closely related to the text,and necessary explanations should be provided for the information presented in graphs or charts.

In the present study, when analyzing the readability of information on different treatment options for breast cancer, we found that the readability of articles in the CAM category was higher for the content, learning stimulation, and motivation dimensions than for the medical and surgical treatment options. Readability was also higher for the cultural appropriateness dimension of CAM than for the comprehensive and medical treatment options. Compared with CAM articles, the medical treatment articles focused too much on the mechanisms of drug action without clearly describing the method of taking the medication and any associated precautions, and provided more information than patients could accept. Additionally, there was a lack of case studies or interactive issues. Doak pointed out that interactive aspects of the material (e.g., asking the reader to answer certain questions, having the reader fill out a self-assessment form, etc.) could increase the reader's sense of engagement in the reading, reinforce motivation to learn, and deepen memory [34]. The cultural appropriateness dimension of the comprehensive treatment articles scored lower than that of the CAM articles, likely because the articles published on WPAs requiring shorter and more concise content [49], while comprehensive treatment focused on the mechanism of action of the treatment when describing two or more treatments, which did not match the literacy level of the readers. Therefore, in the content dimension, when producing medical treatment articles for WPAs, producers attached links to references at the end of the articles for the description of the drug mechanism of action, and add a description of the behavioral information in the body of the article. In the learning stimulation and motivation dimension, medical treatment articles could provide real cases or questions for readers to think and answer, so that readers can interact with the material and improve their interest in reading [34]. In the cultural appropriateness dimension, when producing comprehensive treatment articles, it is recommended to use 3D

modeling and animation videos in the article to explain the mechanism of action of treatment methods, so as to reduce the difficulty of readers to understand professional terms.

In the present study, we compared the readability of articles in different subgroups of medical treatments, and found that the readability of chemotherapy articles was higher than that of clinical trial articles for the content, graphics, and learning stimulation and motivation dimensions, and higher than that for the literacy demand and cultural appropriateness dimensions of targeted therapy articles. According to the circular "Clinical value-oriented clinical development guidelines for antineoplastic drugs" issued by the Chinese Drug Review Center of the State Drug Administration 2021 [50], participation of oncology patients in clinical trials was identified as a means of treatment. One study showed that 25% of patients with cancer used the Internet to find information about clinical trials [51]. However, clinical trial articles provided little information about behavioral skills and applications of the information described, which are precisely what patients are most interested in and want to read [34]. Thirteen (100%) of the articles in the clinical trials category did not use pictures or diagrams, case studies, or a question-and-answer format for interactive design, which may decrease patient interest and motivation [38]. Therefore, the readability rating of the clinical trial articles was "not suitable" for the content, graphics, and learning stimulation and motivation dimensions. As such, it is suggested that the elaboration of behavioral information should be increased when producing clinical trial articles, that complex or comparative information should be presented in a table form, that articles include real cases or raise questions for readers to think about. Compared with the chemotherapy articles, the readability rating of the literacy demand dimension of the targeted therapy articles was "not suitable", largely because the targeted therapy articles used more medical terms without defining them upon their first use, which affected the overall expression of the article and made it harder to understand. In view of this, it is suggested that when producing targeted therapy articles, medical terminology should be minimized or explained when it appears for the first time. And animation could be used to express the mechanism of action of targeted drugs.

In the readability analysis of the articles discussing CAM therapy subgroups in the present study, we found that the readability of complementary therapy articles was higher for the literacy demand and cultural appropriateness dimensions than in alternative therapy articles. The alternative therapy articles often used specialized terminology. And most of the articles on alternative therapies had fewer examples and were purely theoretical, which did not fit with the patients' life experiences. Therefore, it is recommended that more well-known words be used, rather than jargon or terminology, thereby increasing readers' interest in reading [52]. It is also recommended to introduce alternative breast cancer therapies with more examples relevant to patients' life experiences.

The present study had several advantages. First, common search strategies used by the public were mimicked to generate WPAs that were likely to be commonly read by the public. Secondly, two evaluators independently reviewed the webpages and compared the results to reduce subjectivity. Thirdly, breast cancer treatment information was categorized by type to explore the readability profiles and attributes of each treatment type from a new perspective.

On the other hand, the present study also had some limitations. First, our web search was not exhaustive, and the online health information obtained from the search should be more extensive and not only based on idiosyncratic searches. Patients may also use other combinations of search terms that yield relevant results, which may include more comprehensive information about breast cancer treatment options. Secondly, the present study only evaluated online information on breast cancer treatment options presented in Chinese, and the findings, therefore, may not apply to other languages. Thirdly, the articles included in the present study

were limited to WPAs already published on April 11, 2021. It is worth noting that different results are generated additional WPAs are published or updated.

## Conclusion

According to the SAM scale scores, the overall readability of WPA-based health information on different treatment options for breast cancer is "adequate". The readability score of CAM treatment articles was higher than that of medical and comprehensive treatment articles, while medical treatment articles had "not suitable" ratings for the learning motivation and motivation dimensions. We also found that the readability rating of the clinical trial treatment articles in the medical treatment category was "not suitable" in terms of content, graphics, and learning stimulation and motivation dimensions. Publishers of health information on WPAs must understand the factors influencing the readability of the different breast cancer treatment options, and adopt strategies to improve readability to enhance the effectiveness of disseminating the treatment information.

## Supporting information

**S1 File. Basic information of 37 WeChat public accounts and 136 breast cancer treatment articles.**
(XLSX)

**S2 File. The Chinese version of the Suitability Assessment of Materials (SAM).**
(DOCX)

**S3 File. The SAM item and dimension scores of 136 articles based on different treatment options and subgroup.**
(XLSX)

## Acknowledgments

We would like to express our gratitude to all who contributed to the writing of this article.

## Author Contributions

**Data curation:** Min Liu.

**Formal analysis:** Bingyan Li.

**Investigation:** Wenjuan Yang.

**Methodology:** Yuxi Zhang.

**Resources:** Jia Liu.

**Software:** Jia Liu, Yuxi Zhang.

**Supervision:** Lunfang Xie.

**Writing – original draft:** Bingyan Li.

**Writing – review & editing:** Bingyan Li.

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
