## [Decision Letter · Decision Letter 0]

29 May 2023

PONE-D-23-12580Comparative readability of information on different treatment options for breast cancer, based on WeChat public accountsPLOS ONE

Dear Dr. Xie,

Thank you for submitting your manuscript to PLOS ONE. After careful consideration, we feel that it has merit but does not fully meet PLOS ONE’s publication criteria as it currently stands. Therefore, we invite you to submit a revised version of the manuscript that addresses the points raised during the review process.

We look forward to receiving your revised manuscript.

Kind regards,

Nenad Filipovic

Academic Editor

PLOS ONE

Journal Requirements:

2. In your Methods section, please include additional information about your dataset and ensure that you have included a statement specifying whether the collection and analysis method complied with the terms and conditions for the source of the data. Please also clarify if the data collection method involved, or was at risk of, collecting personal health information.

Reviewers' comments:

Reviewer's Responses to Questions

**Comments to the Author**

1. Is the manuscript technically sound, and do the data support the conclusions?

Reviewer #1: Yes

2. Has the statistical analysis been performed appropriately and rigorously? 

Reviewer #1: Yes

3. Have the authors made all data underlying the findings in their manuscript fully available?

Reviewer #1: Yes

4. Is the manuscript presented in an intelligible fashion and written in standard English?

Reviewer #1: No

5. Review Comments to the Author

Reviewer #1: This is an interesting study and I congratulate the authors on this manuscript. It is interesting to note the difference in CAM and non-CAM online information.

There are a small number of changes that I think will enhance this manuscript:

1) Page 9 – end of the page it reads “in worldwide, Approximately 29-57%” – could this be changed to “Worldwide, approximately 29-57%”.

2) Introduction is very thorough, but this information can be presented more succinctly.

3) First paragraph of discussion seems to be repeating what has already been discussed in "Introduction"

4) The discussion is very long. I would recommend shortening the length of the discussion to make it easier to read and follow

6. PLOS authors have the option to publish the peer review history of their article (what does this mean?). If published, this will include your full peer review and any attached files.

Reviewer #1: No

---

## [Author Response · Author response to Decision Letter 0]

13 Jul 2023

Journal Requirements:

1.Please ensure that your manuscript meets PLOS ONE's style requirements, including those for file naming.

Response: We have modificated the manuscript and file names to meet PLOS ONE's style requirements including font, font size, line spacing, fig , etc.

2.In your Methods section, please include additional information about your dataset and ensure that you have included a statement specifying whether the collection and analysis method complied with the terms and conditions for the source of the data. Please also clarify if the data collection method involved, or was at risk of, collecting personal health information.

Response: Thank you for the suggestion. In the methods section, we have added a detailed process for screening WPAs and articles, as described in page 6, line 121-126 and page 7, line 128-136. Basic characteristics information extraction for articles, as described in page 7, line 143-151, and the detailed dataset is also presented as S1 File in the supporting information. The 136 articles included in this study did not involve personal health information and are clarified in the text, as described in page 7, line 136-137. 

3.In your Data Availability statement, you have not specified where the minimal data set underlying the results described in your manuscript can be found. PLOS defines a study's minimal data set as the underlying data used to reach the conclusions drawn in the manuscript and any additional data required to replicate the reported study findings in their entirety. All PLOS journals require that the minimal data set be made fully available.

Response: We gratefully appreciate for your comment. The “Data Availability Statement” have listed before the acknowledgments. Please see the revised content on page 20, line 414-415. The minimal dataset for our study is the readability score of 136 articles, which has been added as the S1 File in the revised supporting information. Please see the revised content on page 26, line 538-541. 

Thanks again for your valuable comment.

Special thanks to your constructive comments.

Responses to Reviewer #1

1: This is an interesting study and I congratulate the authors on this manuscript. It is interesting to note the difference in CAM and non-CAM online information.

Response: We thank you for your appreciation of our efforts. Your comments and advices are of great help to improve our study and manuscript. We have revised the article and made the following modifications based on your comments.

1)Page 9–end of the page it reads “in worldwide, Approximately 29-57%”–could this be changed to “Worldwide, approximately 29-57%”.

Response: Thank you for pointing out this problem in manuscript. we have made the revision on page 4, line 79.

2) Introduction is very thorough, but this information can be presented more succinctly.

Response: Thank you for the detailed review. We have merged and made concise the two paragraphs describing WPAs in the Introduction. Please see the revised content on page 4, line 72-78.

3)First paragraph of discussion seems to be repeating what has already been discussed in "Introduction".

Response: Thank you for pointing this out. We have modified the first paragraph of discussion. Please see the revised content on page 14, line 270-276. 

4) The discussion is very long. I would recommend shortening the length of the discussion to make it easier to read and follow.

Response: Thank you for your suggestion. We reorganized the discussion section, removing duplicates and highlighting the shortcomings and suggestions for improvement of the readability of different treatment options. 

Once again, we appreciate you for your time and effort on revision, and the constructive comments and advices that improve out manuscript.

If the reviewers and editors have questions or are not satisfied with our revisions and responses, please let us know. We are willing to actively modify the text to improve the quality of the article.

---

## [Decision Letter · Decision Letter 1]

25 Sep 2023

PONE-D-23-12580R1Comparative readability of information on different treatment options for breast cancer, based on WeChat public accountsPLOS ONE

Dear Dr. Xie,

Thank you for submitting your manuscript to PLOS ONE. After careful consideration, we feel that it has merit but does not fully meet PLOS ONE’s publication criteria as it currently stands. Therefore, we invite you to submit a revised version of the manuscript that addresses the points raised during the review process.

We look forward to receiving your revised manuscript.

Kind regards,

Nenad Filipovic

Academic Editor

PLOS ONE

Reviewers' comments:

Reviewer's Responses to Questions

**Comments to the Author**

1. If the authors have adequately addressed your comments raised in a previous round of review and you feel that this manuscript is now acceptable for publication, you may indicate that here to bypass the “Comments to the Author” section, enter your conflict of interest statement in the “Confidential to Editor” section, and submit your "Accept" recommendation.

Reviewer #2: All comments have been addressed

Reviewer #3: (No Response)

2. Is the manuscript technically sound, and do the data support the conclusions?

Reviewer #2: Yes

Reviewer #3: No

3. Has the statistical analysis been performed appropriately and rigorously? 

Reviewer #2: Yes

Reviewer #3: No

4. Have the authors made all data underlying the findings in their manuscript fully available?

Reviewer #2: Yes

Reviewer #3: No

5. Is the manuscript presented in an intelligible fashion and written in standard English?

Reviewer #2: Yes

Reviewer #3: Yes

6. Review Comments to the Author

Reviewer #2: Was the readability assessment performed by two physicians? Since this is a general

audience article, why not ask a non-medical person to evaluate it? Would you get a

different result?

Reviewer #3: Thank you for the chance to review this study that calculated readability levels for online information about breast cancer.

The researchers selected 136 articles on various aspects of treatment for breast cancer that were published on 37 different WeChat accounts. Readability is an important to help ensure equity of care irrespective of levels of formal educations.

I appreciate the work that has gone into the paper, including responding to previous review comments.

Unfortunately I had difficulty in understanding the methods and the rationale for the study which lacked many important details and also lacked the source data that I would expect in a such a study, even allowing for the language difficulties.

Data access

Although the researchers say they have provided all relevant material, they only provide the readability scores for these papers and not the papers themselves.

Even though I do not understand Chinese, the researchers have used a tool to evaluate readability that focusses on the visual layout of material, including the use of headings, space, illustrations and other graphics (top of page 8: "graphics, layout and typography, learning stimulation and motivation, and cultural appropriateness”) . These are actually easy to understand irrespective of language, as the analysis or the paper explains. Currently, the results of the study are presented - ie the readability score - but not the actual examples that the study is based on. If this study was about readability of resources produced in English, I would want to look at these original papers which would immediately tell me about the range of resources. This option to see the source material should be available to Chinese readers.

The most important aspect of readability for resources in English is the language used, the structure of the language and the simplicity of the words. At it’s simplest, this involves using short simple words, short sentences and short paragraphs, using the Flecsh-Kincaid or similar reading ease score. Because structural challenges of the Chinese language have not led to a translatable version of Flecsh-Kincaid or similar, test readability has been dropped from the analysis, only relying on visual aspects mentioned above, making access to the source resources essential.

Background and general comments

As an English reader, and as the publication is in English, it is not helpful that the paper has no explanation for WeChat as a publisher. Without this context it is difficult to decide on the context of this information. WeChat appears to be similar to FaceBook, and so-called Official or Public accounts look similar to a Facebook profile for individuals where individuals or organisations can publish content commercial or free content, also building up followers. Content doesn’t appear to need to pass any quality control but is just self-published.

WeChat would be easily known to a Chinese reader - with apparently 360 million users in 2021 according to the paper - so will be a vast source of potential information, but there is little information for how the researcher selected 126 articles or the 37 WeChat profiles. The last paragraph on page 6 just says ’37 WPAs were selected’.

Without, for example, knowing if these are just personal accounts or breast cancer charities/NGOs I have no interest in readab ility scores.

The cut-off criteria for poor readability also seems like an unusually low threshold - ie anything above reading grade 6. Very little general medical information in English achieves reading grade 6 or lower, and yet the main result from the paper is that much of the material they looked at is reading grade 7.

Although ‘content’ of the resources is included in the SAMS algorithm, the paper doesn’t explain how this was evaulated - unless I missed this, and apologise. Yet content is arguably the most important criteria for knowing whether the material is a good source of information. I don’t want to know whether unreliable information is easy to read.

Introduction

Although breast cancer clearly affects a very large numbers of people I worry when I read sentences like:

"Worldwide, 29–57% of breast cancer patients have searched for relevant health information online.” (halfway down page 4).

The context is the rapid development of the internet and this statement is based on two tiny studies (approx 200 people) that were published over a decade ago.

I have no confidence that 66.7% of people with breast cancer in China searched for information online. It is not true and isn;t a fact. It makes be see this paper, unfortunately, as a lot of generated data, that is disconnected from any real-world understanding of the issues related to readability of real information used by real people who depend on this for essential life choices about their future.

Statistics

Perhaps this is a cultural difference that I am not aware of, but given the paper refers to an unequal distribution of data, it is appropriate to present median and IQR for all data. I don’t understand why the IQR is given as a numeral rather than a range - ie being presented as an SD when standard practice ot to give a clear interquartile range.

7. PLOS authors have the option to publish the peer review history of their article (what does this mean?). If published, this will include your full peer review and any attached files.

Reviewer #2: No

Reviewer #3: No

---

## [Author Response · Author response to Decision Letter 1]

7 Nov 2023

Point-by-point response to reviewers

Manuscript ID PONE-D-23-12580R1

Title: Comparative readability of information on different treatment options for breast cancer, based on WeChat public accounts

Dear editors and reviewers,

We appreciate the time and effort that you and the reviewer dedicated to our manuscript. We are also grateful for the insightful comments and valuable suggestions. In the revised manuscript, the changes are marked in the “Track Changes” mode. Below are notes on the revised manuscript, with our answers highlighted in blue. We hope that the revision has addressed the concerns raised by the reviewer, and the revision is now found satisfactory. We are ready to work on resolving any remaining issues.

Sincerely,

Bingyan Li

Corresponding author:

Lunfang Xie

E-mail: 527548725@qq.com

Responses to Reviewer #2

Was the readability assessment performed by two physicians? Since this is a general audience article, why not ask a non-medical person to evaluate it? Would you get a different result?

Response: We gratefully appreciate your comments. 

Q1: Was the readability assessment performed by two physicians? 

A: No. In our study, two postgraduate medical students (BL and WY) used SAM to evaluate the readability of the articles. BL and WY are part of our research team who has been studying on the readability of health information in China in recent years. They are proficient in the use of SAM and have published six articles as belows.

[1] Yang W, Li B, Xie L, et al. Quality evaluation of health information about breast cancer treatment found on WeChat public accounts. Arch Public Health. 2023;81(1):170. Published 2023 Sep 15.

[2] Wang Q, Xie L, Xing L, et al. Development of the readability evaluation index system for online health information. Chinese Journal of Nursing Education 2022;19(01):14-20.

[3] Wang L, Xie L, Wang Q, et al. Readability assessment of online health information for Systemic Lupus Erythematosus. Nursing Research of China 2020;34(24):4344-4349.

[4] Wang L, Xie L, Wang Q, et al. Study Status and Prospect of Readability Assessment of Online Health Information. Journal of Medical Informatics 2020;41(12):20-25+40.

[5] ZhiWei W, Xie L, YuanYuan F, et al. The readability assessment of printed education materials on Hypertension. Chinese Journal of Health Education 2020;36(12):1122-1125+1167.

[6]. Wang Q, Xie L, Li M, et al. Review on readability assessment tools for health education materials. Chinese Journal of Health Education 2019;35:66-71.

[7]. Wang Q, Xie L, Li M, et al. Evaluation of quality and readability of health education materials in Anhui Province from nurses’perspectives. Chinese Journal of Modern Nursing 2019;25:1093-98.

Please see the revised content on page 9, line 186-187. 

Q2: Why not ask a non-medical person to evaluate it? Would you get a different result?

A: The SAM was developed to be used by professionals, not by a non-medical person. According to the literature review, most study on readability assessment of medical information chose medical specialists and medical students as evaluators, because they are more familiar with medical knowledge, they can evaluate the readability based on the accuracy of medical information. If we ask a non-medical person to evaluate, the result could be very different, but not credible.

Responses to Reviewer #3

The researchers selected 136 articles on various aspects of treatment for breast cancer that were published on 37 different WeChat accounts. Readability is an important to help ensure equity of care irrespective of levels of formal educations.

I appreciate the work that has gone into the paper, including responding to previous review comments.

Unfortunately I had difficulty in understanding the methods and the rationale for the study which lacked many important details and also lacked the source data that I would expect in a such a study, even allowing for the language difficulties.

About Data access:

1.Although the researchers say they have provided all relevant material, they only provide the readability scores for these papers and not the papers themselves.

Even though I do not understand Chinese, the researchers have used a tool to evaluate readability that focuses on the visual layout of material, including the use of headings, space, illustrations and other graphics (top of page 8: "graphics, layout and typography, learning stimulation and motivation, and cultural appropriateness”) . These are actually easy to understand irrespective of language, as the analysis or the paper explains. Currently, the results of the study are presented - ie the readability score - but not the actual examples that the study is based on. If this study was about readability of resources produced in English, I would want to look at these original papers which would immediately tell me about the range of resources. This option to see the source material should be available to Chinese readers.

The most important aspect of readability for resources in English is the language used, the structure of the language and the simplicity of the words. At it’s simplest, this involves using short simple words, short sentences and short paragraphs, using the Flecsh-Kincaid or similar reading ease score. Because structural challenges of the Chinese language have not led to a translatable version of Flecsh-Kincaid or similar, test readability has been dropped from the analysis, only relying on visual aspects mentioned above, making access to the source resources essential.

Response: Thanks for bringing up the concern. The basic characterization information for 37 WPAs and 136 articles is listed in S1 file, which is also added in Supporting Information on page 28, lines 579-581. The S1 File supplements the Chinese titles, WPA owners, links to the original articles and treatment options for all the WPAs and articles, so that experts and readers can access to the source resources.

About Background and general comments:

1.As an English reader, and as the publication is in English, it is not helpful that the paper has no explanation for WeChat as a publisher. Without this context it is difficult to decide on the context of this information. WeChat appears to be similar to FaceBook, and so-called Official or Public accounts look similar to a Facebook profile for individuals where individuals or organisations can publish content commercial or free content, also building up followers. Content doesn’t appear to need to pass any quality control but is just self-published.

Response: Based on your suggestions, we have added more background information about WeChat, such as the developers and use of WeChat on page 4, line 73-76, the WPA owners on page 4, line 76-78, and the description of WeChat's qualification of WPA owners and content quality control is on page 5, line 99-104.

2.WeChat would be easily known to a Chinese reader - with apparently 360 million users in 2021 according to the paper - so will be a vast source of potential information, but there is little information for how the researcher selected 136 articles or the 37 WeChat profiles. The last paragraph on page 6 just says ’37 WPAs were selected’.

Response: We gratefully appreciate your comments. We have added more detailed information for how the researcher selected 136 articles or the 37 WeChat profiles, such as the search process for WPAs, the inclusion and exclusion criteria for articles and WPAs, and final compliance with the criteria. Please see the revised content on Page 6, Line 126-134 and Page 7, Line 137-146. 

3.Without, for example, knowing if these are just personal accounts or breast cancer charities/NGOs I have no interest in readability scores.

Response: Thank you for your constructive comment. We've added WPA owners content on page 8, line 156-158 and page 10, line 211-212. The information for 37 WPA owners is listed in S1 file in Supporting Information on page 28, lines 579-581. 

4.The cut-off criteria for poor readability also seems like an unusually low threshold - ie anything above reading grade 6. Very little general medical information in English achieves reading grade 6 or lower, and yet the main result from the paper is that much of the material they looked at is reading grade 7.

Response: Grade 6 is calculated based on the formula, which mainly calculates the difficulty level of word comprehension from the perspective of textual factor, while the readability evaluation in this study is based on six dimensions of SAM from the perspective of non-textual factors. So “adequate” in our study might not be equal to grade 7.

5.Although ‘content’ of the resources is included in the SAM algorithm, the paper doesn’t explain how this was evaulated - unless I missed this, and apologise. Yet content is arguably the most important criteria for knowing whether the material is a good source of information. I don’t want to know whether unreliable information is easy to read.

Response: 

(1)The content dimension in the SAM consists of 4 factors, as described in the following table.

Content Dimension

(a) Purpose is evident

2 Purpose is explicitly stated in title, or cover illustration, or introduction

1 Purpose is not explicitly. It is implied, or multiple purposes are stated

0 No purpose is stated in the title cover illustration, or introduction

(b) Content about behaviors

2 Thrust of the material is application of knowledge/skills aimed at

1 Desirable reader behavior rather than non-behavior facts

0 Nearly all topics are focused on non-behavior facts

(c) Scope is limited

2 Scope is limited to essential information directly related to the purpose. Experience shows it can be learned in time allowed.

1 Scope is expanded beyond the purpose; no more than 40 percent is non-essential information. Key reports can be learned in time allowed

0 Scope is far out of proportion to the purpose and time allowed

(d) Summary or review included

2 A summary is included and retells the key messages in different words and examples

1 Some key ideas are reviewed.

0 No summary or review is included

 The evaluation process for the content dimension is as same as that for the other dimensions. We have added relevant content on improving inter-rater consistency and the assessment process. Please see the revised content on page 9, line 186-194. 

(2)This article is part of our study named ‘quality and readability evaluation on breast cancer treatment information on WPAs’, and anther paper about the quality of informaition has been published with findings of moderate quality, cf. [33]. In particular, DISCERN which consists of three parts and 16 items was used to assess quality of 136 articles. The first part (Items 1–8) is related to the reliability of the article; the second part (Items 9–15) evaluates the specific details of treatment choice information; the third part (Item 16) describes the overall quality of the source publication of health information on treatment choices. The reliability dimension was assessed as fair, with criteria including: clear information sources, explicit objectives, relevant content themes, objective content, and mention of uncertainties. Please see the revised content on page 6, line 120-122. 

About Introduction:

1.Although breast cancer clearly affects a very large numbers of people I worry when I read sentences like:

"Worldwide, 29–57% of breast cancer patients have searched for relevant health information online.” (halfway down page 4).

The context is the rapid development of the internet and this statement is based on two tiny studies (approx 200 people) that were published over a decade ago.

Response:Thank you for the detailed review. The statement you mentioned has been updated with new references. Please see the revised content on page 4, line 72-73.

2.I have no confidence that 66.7% of people with breast cancer in China searched for information online. It is not true and isn’t a fact. It makes be see this paper, unfortunately, as a lot of generated data, that is disconnected from any real-world understanding of the issues related to readability of real information used by real people who depend on this for essential life choices about their future.

Response:Thank you for your suggestion. We have rephrased the use of WPAs in breast cancer patients, as well as added the sources from which this conclusion was reached to add credibility, and added some literatures to support the data. See revisions to lines 83-86 on page 4.

About Statistics:

1.Perhaps this is a cultural difference that I am not aware of, but given the paper refers to an unequal distribution of data, it is appropriate to present median and IQR for all data. I don’t understand why the IQR is given as a numeral rather than a range - ie being presented as an SD when standard practice ot to give a clear interquartile range.

Response: In our understanding, the IQR can be either a range or the difference between P75 and P25, cf. [36]. Now, we have expressed IQR as a range based on your suggestion. See Table 1 on page 11, Table 2 on page 13, and Table 3 on page 14 for revisions, and the representation of IQR in the text has also been revised (see Page 2, Line 28-33 and Page 11, Line 222). 

Once again, we appreciate you for your time and effort on revision, and the constructive comments and advice that improve out manuscript.

If the reviewers and editors have questions or are not satisfied with our revisions and responses, please let us know. We are willing to actively modify the text to improve the quality of the article.

---

## [Decision Letter · Decision Letter 2]

26 Mar 2024

PONE-D-23-12580R2Comparative readability of information on different treatment options for breast cancer, based on WeChat public accountsPLOS ONE

Dear Dr. Xie,

Thank you for submitting your manuscript to PLOS ONE. After careful consideration, we feel that it has merit but does not fully meet PLOS ONE’s publication criteria as it currently stands. Therefore, we invite you to submit a revised version of the manuscript that addresses the points raised during the review process. Please submit your revised manuscript by May 10 2024 11:59PM. If you will need more time than this to complete your revisions, please reply to this message or contact the journal office at plosone@plos.org. Please include the following items when submitting your revised manuscript:A rebuttal letter that responds to each point raised by the academic editor and reviewer(s). You should upload this letter as a separate file labeled 'Response to Reviewers'.A marked-up copy of your manuscript that highlights changes made to the original version. You should upload this as a separate file labeled 'Revised Manuscript with Track Changes'.An unmarked version of your revised paper without tracked changes. You should upload this as a separate file labeled 'Manuscript'.If applicable, we recommend that you deposit your laboratory protocols in protocols.io to enhance the reproducibility of your results. Protocols.io assigns your protocol its own identifier (DOI) so that it can be cited independently in the future. For instructions see: https://journals.plos.org/plosone/s/submission-guidelines#loc-laboratory-protocols. Additionally, PLOS ONE offers an option for publishing peer-reviewed Lab Protocol articles, which describe protocols hosted on protocols.io. Read more information on sharing protocols at https://plos.org/protocols?utm_medium=editorial-email&utm_source=authorletters&utm_campaign=protocols.

We look forward to receiving your revised manuscript.

Kind regards,

Shahabedin Rahmatizadeh, Ph.D.

Academic Editor

PLOS ONE

Journal Requirements:

Additional Editor Comments:

"Thank you for taking the reviewers' comments into consideration and making corrections in the manuscript. Please also review and address the other comments provided by the esteemed reviewer and resubmit the revised manuscript."

Reviewers' comments:

Reviewer's Responses to Questions

**Comments to the Author**

1. If the authors have adequately addressed your comments raised in a previous round of review and you feel that this manuscript is now acceptable for publication, you may indicate that here to bypass the “Comments to the Author” section, enter your conflict of interest statement in the “Confidential to Editor” section, and submit your "Accept" recommendation.

Reviewer #3: All comments have been addressed

Reviewer #4: (No Response)

2. Is the manuscript technically sound, and do the data support the conclusions?

Reviewer #3: Yes

Reviewer #4: Partly

3. Has the statistical analysis been performed appropriately and rigorously? 

Reviewer #3: I Don't Know

Reviewer #4: Yes

4. Have the authors made all data underlying the findings in their manuscript fully available?

Reviewer #3: Yes

Reviewer #4: Yes

5. Is the manuscript presented in an intelligible fashion and written in standard English?

Reviewer #3: Yes

Reviewer #4: Yes

6. Review Comments to the Author

Reviewer #3: Thank you for responding to my previous points - good luck with this and future reseaerch.

Reviewer #4: This is a fascinating study exploring a tool of which I was unaware. WeChat and its WPAs seem like a popular resource for patients seeking reliable medical information, and I applaud your recommendations for making them easier to use and understand.

I do have a few questions, though, about your methods.

In Section 2.2.1 you mention that a translation of the Suitability Assessment of Materials (SAM) was used to gauge the readability of each article. As you note the SAM recommends using the Flesch-Kincaid readability formula to assess the Literacy Demand of an article, and you also note that this particular tool has not been translated into Chinese and therefore this component was deleted.

How, then, was the Literacy Demand calculated? We see a score for this component in each of the Tables, but I’m unsure how this was determined. Did you use the original English version of Flesch-Kincaid, or did you substitute something else? Did you use the Chinese Readability Index Explorer mentioned earlier in the Introduction? I re-read the paper a few times and this escaped me, and I apologize if I simply missed it.

As has been noted by other Reviewers, the Discussion section is too long, and parts of it seem speculative as opposed to rooted in fact. For example, on Page 18, you discuss the benefits of using graphics to support concepts discussed in the text, and you provide a relevant citation. This is well done. However, on Page 19, you mention that patients’ interest level in CAM articles is due to including the topic of nutritional therapy and mentioning “commonly used foods, such as oranges and spinach”. Where is the citation for that? How did you determine interest? Does WeChat display the number of reads for a WPA? Even if you had such usage data, how can you determine which aspect of the paper engendered interest? Does WeChat allow users to post comments, and, if so, did you read them? Or did you read a paper that somehow tracks patient interest and simply forgot to cite it here?

A similar question is raised in the next paragraph where you state,” medical treatment articles tended to lack descriptions of behavioral information that patients valued”. Again, how do you determine what patients value or what they like? Scoring papers on readability or suitability is logical, based on objective, known assessment scales, but determining reasons for popularity is not.

On Page 21 you discuss a circular “Clinical value-oriented clinical development guidelines for antineoplastic drugs” issued by the Chinese Drug Review Center of the State Drug Administration. Is this a WPA? If so, was this written as a WPA, or was it simply posted to WeChat? This passage does a thorough job of articulating the perceived deficiencies of the circular (and again, we are told what patients want with no context or citation, “clinical trial articles provided little information about behavioral skills and applications of the information described, which are precisely what patients are most interested in and want to read”), but if this circular isn’t actually a WPA is this relevant?

My final question has to do with what you hope to achieve with this study, or, perhaps more importantly, how you plan on carrying out this plan. On Page 11 you state, “The present study, therefore, assessed the readability of information available in WPAs regarding different treatment options for breast cancer to clarify the dimensions of relatively high and low readability of each type of information, and to help WPA producers focus on improving the readability of this information”. This is an admirable goal – the world is awash in misinformation, especially on social media platforms discussing medical topics. What is your plan for sharing this study with WeChat? Do you plan on sending the Editorial Board a link to the final, published paper? Are you hoping that WeChat content creators will come across this paper on their own? As a peer reviewer I was taught to always ask a final question about generalizability or ease of implementation, and I don’t know you intend to transmit your findings to your intended audience so that your recommendations can be implemented.

7. PLOS authors have the option to publish the peer review history of their article (what does this mean?). If published, this will include your full peer review and any attached files.

Reviewer #3: No

Reviewer #4: No

---

## [Author Response · Author response to Decision Letter 2]

7 May 2024

Responses to Reviewer #4

This is a fascinating study exploring a tool of which I was unaware. WeChat and its WPAs seem like a popular resource for patients seeking reliable medical information, and I applaud your recommendations for making them easier to use and understand.I do have a few questions, though, about your methods.

About Data access:

1.In Section 2.2.1 you mention that a translation of the Suitability Assessment of Materials (SAM) was used to gauge the readability of each article. As you note the SAM recommends using the Flesch-Kincaid readability formula to assess the Literacy Demand of an article, and you also note that this particular tool has not been translated into Chinese and therefore this component was deleted. How, then, was the Literacy Demand calculated? We see a score for this component in each of the Tables, but I’m unsure how this was determined. Did you use the original English version of Flesch-Kincaid, or did you substitute something else? Did you use the Chinese Readability Index Explorer mentioned earlier in the Introduction? I re-read the paper a few times and this escaped me, and I apologize if I simply missed it.

Response: We gratefully appreciate your comments. 

Q1: How was the Literacy Demand calculated ? Did you use the original English version of Flesch-Kincaid, or did you substitute something else? Did you use the Chinese Readability Index Explorer mentioned earlier in the Introduction?

A: We did not use the original English version of Flesch-Kincaid or other substitute tool. We did not use the Chinese Readability Index Explorer either. In our study, the SAM referred to the Chinese translation by Chinese scholar Xianwen Li in 2013 was used. “Literacy Demand Dimension” in the English version has five items including Reading grade level, Writing style, Vocabulary uses common words, Context is given first, and Learning aids. Reading grade level was assessed by the Flesch-Kincaid readability formula. But this item was not in the Chinese version of the SAM. So we assessed the literacy demand with the rest 4 items. In order to clarify this issue, the article had been revised accordingly. See page 8, line 172-177 and page 9, line 180-185. The dimension and item details of the Chinese version of the Suitability Assessment of Materials (SAM), which had been added as the S2 File in the revised supporting information. Please see the revised content on page 28, line 591-592. The changes in the current article are also reflected, see page 9, line 188-189.

2.As has been noted by other Reviewers, the Discussion section is too long, and parts of it seem speculative as opposed to rooted in fact. For example, on Page 18, you discuss the benefits of using graphics to support concepts discussed in the text, and you provide a relevant citation. This is well done. However, on Page 19, you mention that patients’ interest level in CAM articles is due to including the topic of nutritional therapy and mentioning “commonly used foods, such as oranges and spinach”. Where is the citation for that? How did you determine interest? Does WeChat display the number of reads for a WPA? Even if you had such usage data, how can you determine which aspect of the paper engendered interest? Does WeChat allow users to post comments, and, if so, did you read them? Or did you read a paper that somehow tracks patient interest and simply forgot to cite it here?

Response: We gratefully appreciate your comments. 

Q1: As has been noted by other Reviewers, the Discussion section is too long.

A: We have reorganized the discussion section to make the shortcomings of the different treatment options and suggestions for improving readability more concise.

Q2: However, on Page 19, you mention that patients’ interest level in CAM articles is due to including the topic of nutritional therapy and mentioning “commonly used foods, such as oranges and spinach”. Where is the citation for that? How did you determine interest? Does WeChat display the number of reads for a WPA? Even if you had such usage data, how can you determine which aspect of the paper engendered interest? Does WeChat allow users to post comments, and, if so, did you read them? Or did you read a paper that somehow tracks patient interest and simply forgot to cite it here?

A: We determined that patients were interested in nutrition therapy for CAM articles according to the reference [46].The researchers investigated 1686 articles on "Breast cancer Mutual support circle" published in the WPA and tracked the reading feedback of 148 WPA users. Because the articles in the WPA allow users to like, comment, collect, forward and share, so the researchers inferred users' interests based on the reads, retweets and feedback from users. The results showed that the articles focusing on nutritional therapy and postoperative behavior guidance were more liked and "impressed" by the users. Please see the revised content on page 16, line 321-324. 

3.A similar question is raised in the next paragraph where you state,” medical treatment articles tended to lack descriptions of behavioral information that patients valued”. Again, how do you determine what patients value or what they like? Scoring papers on readability or suitability is logical, based on objective, known assessment scales, but determining reasons for popularity is not.

Response: Thank you for your constructive comment. The original expression was not appropriate and has been revised. See page 17, line 345-352.

4.On Page 21 you discuss a circular “Clinical value-oriented clinical development guidelines for antineoplastic drugs” issued by the Chinese Drug Review Center of the State Drug Administration. Is this a WPA? If so, was this written as a WPA, or was it simply posted to WeChat? This passage does a thorough job of articulating the perceived deficiencies of the circular (and again, we are told what patients want with no context or citation, “clinical trial articles provided little information about behavioral skills and applications of the information described, which are precisely what patients are most interested in and want to read”), but if this circular isn’t actually a WPA is this relevant? 

Response: Thank you for the detailed review.

Q1: A circular “Clinical value-oriented clinical development guidelines for antineoplastic drugs” issued by the Chinese Drug Review Center of the State Drug Administration. Is this a WPA? If so, was this written as a WPA, or was it simply posted to WeChat?

A: The Guidelines for “Clinical value-oriented clinical development guidelines for antineoplastic drugs” is an official implementation guideline for clinical trials, which is published on the organization's official website and not on WPA. We mentioned this guideline here was to explain why the clinical trials was evaluated as one of the categories of treatment options in this paper. 

Q2: This passage does a thorough job of articulating the perceived deficiencies of the circular (and again, we are told what patients want with no context or citation, “clinical trial articles provided little information about behavioral skills and applications of the information described, which are precisely what patients are most interested in and want to read”).

A: We've added references[34], and the revised content was on page 19, line 379-381.

Q3: If this circular isn’t actually a WPA, is this relevant? 

A: Please see the answer to Q1 above.

5.My final question has to do with what you hope to achieve with this study, or, perhaps more importantly, how you plan on carrying out this plan. On Page 11 you state, “The present study, therefore, assessed the readability of information available in WPAs regarding different treatment options for breast cancer to clarify the dimensions of relatively high and low readability of each type of information, and to help WPA producers focus on improving the readability of this information”. This is an admirable goal – the world is awash in misinformation, especially on social media platforms discussing medical topics. What is your plan for sharing this study with WeChat? Do you plan on sending the Editorial Board a link to the final, published paper? Are you hoping that WeChat content creators will come across this paper on their own? As a peer reviewer I was taught to always ask a final question about generalizability or ease of implementation, and I don’t know you intend to transmit your findings to your intended audience so that your recommendations can be implemented.

Response: Thanks for bringing up the concern. Firstly, we hope that this article will be read by the researchers and practitioners in the field of health information communication or health education. They will apply the findings to their practice or future researches. Secondly, we can submit the related articles to the WPAs that focused on breast cancer treatment and spread our findings or suggestions through these platforms. Finally, we will participate in some academic conferences to disseminate this study to article producers or publishers who are providing medical care for breast cancer patients. So we believe that if the paper can be published, the recommendations will be implemented in future.

Once again, we appreciate you for your time and effort on revision, and the constructive comments and advice that improve out manuscript.

If the reviewers and editors have questions or are not satisfied with our revisions and responses, please let us know. We are willing to actively modify the text to improve the quality of the article.

---

## [Decision Letter · Decision Letter 3]

4 Oct 2024

PONE-D-23-12580R3Comparative readability of information on different treatment options for breast cancer, based on WeChat public accountsPLOS ONE

Dear Dr. Xie,

Thank you for submitting your manuscript to PLOS ONE. After careful consideration, we feel that it has merit but does not fully meet PLOS ONE’s publication criteria as it currently stands. Therefore, we invite you to submit a revised version of the manuscript that addresses the points raised during the review process.

We look forward to receiving your revised manuscript.

Kind regards,

Shahabedin Rahmatizadeh, Ph.D.

Academic Editor

PLOS ONE

Journal Requirements:

Additional Editor Comments:

Thank you for revising the article. Considering the reviewers' comments, please address the mentioned issues and submit the revised version to the journal.

Reviewers' comments:

Reviewer's Responses to Questions

**Comments to the Author**

1. If the authors have adequately addressed your comments raised in a previous round of review and you feel that this manuscript is now acceptable for publication, you may indicate that here to bypass the “Comments to the Author” section, enter your conflict of interest statement in the “Confidential to Editor” section, and submit your "Accept" recommendation.

Reviewer #5: (No Response)

Reviewer #6: (No Response)

Reviewer #7: (No Response)

2. Is the manuscript technically sound, and do the data support the conclusions?

Reviewer #5: No

Reviewer #6: (No Response)

Reviewer #7: Yes

3. Has the statistical analysis been performed appropriately and rigorously? 

Reviewer #5: Yes

Reviewer #6: (No Response)

Reviewer #7: Yes

4. Have the authors made all data underlying the findings in their manuscript fully available?

Reviewer #5: Yes

Reviewer #6: (No Response)

Reviewer #7: Yes

5. Is the manuscript presented in an intelligible fashion and written in standard English?

Reviewer #5: Yes

Reviewer #6: (No Response)

Reviewer #7: Yes

6. Review Comments to the Author

Reviewer #5: Articles on breast cancer treatment options found in WeChat are not necessarily the result of research by authors in their specialty, are not peer-reviewed, and have no real experimental data to offer. So these articles are not credible. There is no academic value in studying these articles.

Reviewer #6: (No Response)

Reviewer #7: The article is scientific and systematically organized; even so, it might be improved with the following additions.

7. PLOS authors have the option to publish the peer review history of their article (what does this mean?). If published, this will include your full peer review and any attached files.

Reviewer #5: No

Reviewer #6: No

Reviewer #7: No

---

## [Author Response · Author response to Decision Letter 3]

16 Nov 2024

Point-by-point response to reviewer

Manuscript ID PONE-D-23-12580R1

Title: Comparative readability of information on different treatment options for breast cancer, based on WeChat public accounts

Dear editors and reviewers,

We appreciate the time and effort that you and the reviewer dedicated to our manuscript. We are also grateful for the insightful comments and valuable suggestions. In the revised manuscript, the changes are marked in the “Track Changes” mode. Below are notes on the revised manuscript, with our answers highlighted in blue. We hope that the revision has addressed the concerns raised by the reviewer, and the revision is now found satisfactory. We are ready to work on resolving any remaining issues.

Sincerely,

Bingyan Li

Corresponding author:

Lunfang Xie

E-mail: 527548725@qq.com

Responses to Reviewer #5

1.Articles on breast cancer treatment options found in WeChat are not necessarily the result of research by authors in their specialty, are not peer-reviewed, and have no real experimental data to offer. So these articles are not credible. There is no academic value in studying these articles.

Response: We gratefully appreciate your comments.

(1)National Health Commission of China encourages medical staff to use wechat official accounts and other new media to disseminate health knowledge to improve the public's health knowledge reserve. Nurses, as an important part of medical staff, can provide the public with understandable and useful health information, which will play a role in improving their health literacy. Studies have shown that health wechat public account has become an important way for breast cancer patients to obtain treatment information, but no study has evaluated and compared the readability level of different treatment types of breast cancer information in wechat official accounts. In view of this, health related professionals such as nursing staff should evaluate and understand the real level of readability of breast cancer treatment information in wechat public accounts, in order to provide a basis for designing and selecting highly readable breast cancer treatment information, and improve the effect of health education. 

(2) This article is part of our study named “quality and readability evaluation on breast cancer treatment information on WPAs”, and anther paper about the quality of informaition has been published. Among the 37 WPAs, individual users accounted for the largest share (20, 54%). To ensure the authenticity and security of WPAs, Tencent provides authentication services for WPAs. The authentication information and the WeChat Authentication Unique Identity are displayed in the authentication details of the account. The median of overall quality of 136 articles was 44 (interquartile range = 10.75) and ranked as "fair", of which 28 (21%) were of "good" or higher quality. cf. [33].Yang W, Li B, Liu M, et al. Quality evaluation of health information about breast cancer treatment found on WeChat public accounts. Arch Public Health. 2023;81(1):170. Published 2023 Sep 15. In particular, DISCERN which consists of three parts and 16 items was used to assess quality of 136 articles. The first part (Items 1–8) is related to the reliability of the article; the second part (Items 9–15) evaluates the specific details of treatment choice information; the third part (Item 16) describes the overall quality of the source publication of health information on treatment choices. The reliability dimension was assessed as fair, with criteria including: clear information sources, explicit objectives, relevant content themes, objective content, and mention of uncertainties. Please see the revised content on page 5, line 107-109. 

(3) In our study, two postgraduate medical students (BL and WY) used SAM to evaluate the readability of the articles. BL and WY are part of our research team who has been studying on the readability of health information in China in recent years. The following articles are the relevant achievements of our team in recent years.

[1] Yang W, Li B, Xie L, et al. Quality evaluation of health information about breast cancer treatment found on WeChat public accounts. Arch Public Health. 2023;81(1):170. Published 2023 Sep 15.

[2] Wang Q, Xie L, Xing L, et al. Development of the readability evaluation index system for online health information. Chinese Journal of Nursing Education 2022;19(01):14-20.

[3] Wang L, Xie L, Wang Q, et al. Readability assessment of online health information for Systemic Lupus Erythematosus. Nursing Research of China 2020;34(24):4344-4349.

[4] Wang L, Xie L, Wang Q, et al. Study Status and Prospect of Readability Assessment of Online Health Information. Journal of Medical Informatics 2020;41(12):20-25+40.

[5] ZhiWei W, Xie L, YuanYuan F, et al. The readability assessment of printed education materials on Hypertension. Chinese Journal of Health Education 2020;36(12):1122-1125+1167.

[6] Wang Q, Xie L, Li M, et al. Review on readability assessment tools for health education materials. Chinese Journal of Health Education 2019;35:66-71.

[7] Wang Q, Xie L, Li M, et al. Evaluation of quality and readability of health education materials in Anhui Province from nurses’perspectives. Chinese Journal of Modern Nursing 2019;25:1093-98.

[8] Li B, Liu M, Liu J, Zhang Y, Yang W, Xie L. Quality assessment of health science-related short videos on TikTok: A scoping review. Int J Med Inform. 2024 Jun;186:105426. 

[9] Liu M, Zhou Y, Li B. Research progress of health science video quality evaluation tools. Chinese General Practice. DOI: 10.12114/j.issn.1007-9572.2023.0683.

 Please see the revised content on page 8, line 173-174. 

Responses to Reviewer #7

1.The SAM tool has been utilized in measuring the readable level of breast cancer treatment information that can be found on WeChat. Through the use of CAM articles, the readers got to see that the readability was higher in comparison to those in medicine and surgery. The clinical trials were disapproved of as the weakest part of the content and graphics. The authors advise the usage of more images and other related interactive elements to exhort the right readers to read the articles. It is a good article with a clear structure and the correct order of the sections. The article was easy to the eyes, and the points were well developed but the title is too long. The main problem is candidly communicated and the area of emphasis is chosen by an example.

Response: We gratefully appreciate your comments. 

(1)The purpose of this study is to evaluate the readability of articles on different treatment methods of breast cancer in WPAs. We tried to modify the title, but it was found that the revised title did not reflect the research focus in detail, so the title is not modified for the time being after discussion by the research team. If the experts have good advice to offer, we will consider it carefully.

(2)In the discussion section, in order to fully reflect the application value of the results of this study, our team not only analyzed each dimension of SAM, but also analyzed each item and made personalized suggestions accordingly. Referring to the expert's parsimonious advice, we have only condensed the questions, but not the recommendations. Please see the revised content on page 16, line 307-311. 

2.The abstract is laid out skillfully and follows a clear sequence of thoughts. The introduction, firstly, provides the research context and its significance in a well-constructed manner, so the readers are sure of the urgency of the issue (readability of health information) informing the most recent statistics and data. Nevertheless, the introduction might be more focused and relay only the scientific implications of the study. The methods are unambiguously and completely explained with a perfect description of the sampling, the (SAM) instrument used as well as the statistical analysis. 

Response: Thank you for pointing this out. Based on your suggestion, the contents of the introduction have been briefly expressed, focusing on the research significance. Please see the revised content on page 4, line 73-84 and page 5, line 97-109 .

3.The findings that are tabulated and charted in the best way to make them more comprehensible are exhibited. The discussion is a deep analysis of the results and elucidates on the points for the differences between the groups. In general, the article is scientific and systematically organized; even so, it might be improved with the following additions. "Is the sole use of the SAM tool sufficient for assessing readability? Why have other multidimensional criteria, such as the use of language complexity models or natural language processing (NLP) analyses, not been employed to examine the complexity of sentences and language structures in these articles? Can multidimensional assessments provide more comprehensive results?"

Response: We gratefully appreciate for your comment. This study evaluated the readability of articles on different treatments of breast cancer in WPAs from the perspective of non-text factors, and therefore did not use the language complexity models or natural language processing (NLP) . On the other hand, there is a dimension of "literacy needs" in SAM, and the words and phrases of articles were also briefly evaluated. The assessment items included: (1) Writing style, active voice; (2) Vocabulary uses common words. 

4.For the reason that of the study is confined to Chinese WeChat articles, how do we ensure the absence of cultural bias in the readability analysis.

Response: Thanks for bringing up the concern. The SAM was translated into Chinese by Li Xianwen in 2010[1]. Since then, the SAM scale has been widely used to evaluate the readability of health information in China, verifying the applicability of the scale in evaluating the readability of health information in China. In addition, after obtaining the authorization of the English version of SAM, the research group compiled the SAM handbook from the perspective of Chinese users by referring to the Chinese version of SAM by Li Xianwen. Each item was explained in detail and examples were provided to avoid cultural bias due to possible differences in the understanding of each item between Chinese and foreigners. In order to verify the comprehensibility of the SAM handbook and the accurate expression of SAM items, we evaluated the readability of SLE health information in 2020[2]. At the same time, the views of SLE patients with high and low health literacy on readability were collected through personal interviews to verify the applicability of the SAM handbook in evaluating the readability of health information in China. Therefore, the assessors of this study used the validated SAM handbook to assess the readability of breast cancer treatment information in the wpa to reduce cultural bias in the assessment process. Please see the revised content on page 8, line 173-174.

References:

[1] Xianwen Li. Suitability evaluation of the health education text material and the health literacy intervention of Korean-Chinese older adults with hypertension. [D] Yanjie: Yanbian University, 2013.

[2] Q Wang. Readability assessment of printed health education materials regarding Systemic Lupus Erythematosus[D].Anhui Medical University,2021.DOI:10.26921/d.cnki.ganyu.2020.001063.

5."Have the educational level and prior knowledge of the patients been taken into consideration in the readability analysis of the articles? How do we make sure that the reviewed articles are suitable for a diverse audience with different health literacy levels?"

Response: Thank you for pointing this out. 

The score of SAM was "superior", indicating that the health information could be understood by the target population with low health literacy[1]. When SAM score is "adequate", it means that health information can be understood by the target population with high health literacy[2]. The SAM score of "not suitable" indicates that the health information cannot be understood by the target population[3].

There is no assessment of education level and prior knowledge involved in the impression, and the traditional assessment results are understandable to most people. However, no research reports correspond to specific grades, unless the English readability assessment formula is used, which is only evaluated from the perspective of words, without considering other important dimensions, and the evaluation of readability is not comprehensive enough.

References:

[1]Tzeng YF, Gau BS. Suitability of asthma education materials for school-age children: Implications for health literacy. J Clin Nurs. 2018 Mar;27(5-6):e921-e930. 

[2]Taylor-Clarke K, Henry-Okafor Q, Murphy C, Keyes M, Rothman R, Churchwell A, Mensah GA, Sawyer D, Sampson UK. Assessment of commonly available education materials in heart failure clinics. J Cardiovasc Nurs. 2012 Nov-Dec;27(6):485-94.

[3]Martin CA, Khan S, Lee R, Do AT, Sridhar J, Crowell EL, Bowden EC. Readability and Suitability of Online Patient Education Materials for Glaucoma. Ophthalmol Glaucoma. 2022 Sep-Oct;5(5):525-530.

6.What ways can we propose to make the articles more comprehensible for the audiences with the lower literacy levels?

Response: Thank you for your constructive comment. When the SAM score is "superior", it means that the health information can be understood by the target population with low health literacy. When the SAM score is "adequate" or "not suitable", it means that the health information cannot be understood by the target population with low health literacy. Therefore, when the SAM score is "adequate" or "not suitable", the low literacy target population must be valued. In this study, the SAM scale was used to evaluate and compare the readability of articles on different treatment methods for breast cancer in WPAs, and the results showed that the overall readability was "adequate". Suggestions were made based on items with low SAM scores, see page 16, line 307-311 for details. Recommendations for medical and comprehensive care articles are provided on page 17 and 18, line 329-339. "We also subcategorized articles on medical treatment and CAM treatment, evaluated and compared their readability, and made recommendations for improvement based on readability results, as detailed on page 18, line 356-366 and page 19, 373-376."

7.Taking into account the fact that a major issue highlighted in the analyzed articles is the lack of pictures and illustrations that go along with the written content, what types of visuals might be the ones to improve the clarity and the comprehension?

Response: Thanks for bringing up the concern.The "graphic" dimension of the SAM scale focuses on the illustration of the article, also known as the pictures. Combining illustration and text can improve the readability of information. Here are some examples and applications of illustrations.

(1)Explanatory illustration: its role is to assist the text, to achieve a visual effect. For example, some health information will involve some acupoints and various movements of the human body. It is necessary to draw some "illustrations" to assist the explanation, so that the audience can see the specific location at a glance.

(2)Embellished illustration: their role is to liven up the page and fill in the blanks. This kind of illustration is used to modify the layout effect. No strict text content requirements, flexibility is strong. It can be used to beautify the title, fill in the gaps where the text is not enough, and can also be used to distinguish the structure of the text block.

(3)Thematic illustration: its role is to deepen the theme, show the theme, publicize the theme. Health information may change with the change of season and epidemic season, so each issue has its theme, and each page has its theme content. Therefore, the illustrations on the layout should also be conceived and expressed according to different themes. Such illustrations can not only serve as a foil, serve between the words, but also can be independent into paintings, creating a distinct feeling effect. Present a theme with text on the same page.

Once again, we appreciate you for your time and effort on revision, and the constructive comments and advice that improve out manuscri

---

## [Decision Letter · Decision Letter 4]

20 Dec 2024

Comparative readability of information on different treatment options for breast cancer, based on WeChat public accounts

PONE-D-23-12580R4

Dear Dr. lunfang Xie,

We’re pleased to inform you that your manuscript has been judged scientifically suitable for publication and will be formally accepted for publication once it meets all outstanding technical requirements.

Kind regards,

Shahabedin Rahmatizadeh, Ph.D.

Academic Editor

PLOS ONE

Additional Editor Comments (optional):

Thank you for addressing the comments made by the reviewers. A very good paper has been prepared, and I hope it will attract the attention of the journal's readers.

Reviewers' comments:

Reviewer's Responses to Questions

**Comments to the Author**

1. If the authors have adequately addressed your comments raised in a previous round of review and you feel that this manuscript is now acceptable for publication, you may indicate that here to bypass the “Comments to the Author” section, enter your conflict of interest statement in the “Confidential to Editor” section, and submit your "Accept" recommendation.

Reviewer #8: All comments have been addressed

Reviewer #9: All comments have been addressed

2. Is the manuscript technically sound, and do the data support the conclusions?

Reviewer #8: Yes

Reviewer #9: Yes

3. Has the statistical analysis been performed appropriately and rigorously? 

Reviewer #8: Yes

Reviewer #9: Yes

4. Have the authors made all data underlying the findings in their manuscript fully available?

Reviewer #8: Yes

Reviewer #9: Yes

5. Is the manuscript presented in an intelligible fashion and written in standard English?

Reviewer #8: Yes

Reviewer #9: Yes

6. Review Comments to the Author

Reviewer #8: This is a very interesting study that evaluates and compares the readability of breast cancer treatment information on public WeChat accounts, confirming that the readability of articles on medical treatment options is poor and has significant room for improvement.

This research has already been evaluated previously. The authors have made a great effort by applying the majority of the proposed recommendations, thus substantially improving their paper.

In this regard, they have enhanced the arguments for the credibility of WeChat by providing an in-depth description of WPAs. They also better justify the added value of their research and how it can serve as a powerful tool to improve health education.

Additionally, they have expanded significantly on the Suitability Assessment of Materials (SAM) and its value as a tool for evaluating both textual and non-textual complexity.

They have also provided much more detail on how they adapted the SAM to the Chinese context, using specific translations and guidelines for this purpose.

Moreover, they simplified the text and analyzed each dimension of the SAM, offering specific recommendations to improve readability.

Finally, while it is true that they decided not to change the title, this reviewer considers this decision legitimate, and their justification is reasonable.

Considering all the effort put into improving the manuscript, this reviewer believes that no further adjustments are necessary and that this research deserves to be published.

Reviewer #9: I reviewed this research and thought it to be an intriguing study, and I applaud the writers on this manuscript. It is fascinating to compare CAM and non-CAM web information.

7. PLOS authors have the option to publish the peer review history of their article (what does this mean?). If published, this will include your full peer review and any attached files.

Reviewer #8: No

Reviewer #9: No

---

## [Editor Report · Acceptance letter]

13 Jan 2025

PONE-D-23-12580R4 

PLOS ONE

Dear Dr. Xie, 

I'm pleased to inform you that your manuscript has been deemed suitable for publication in PLOS ONE. Congratulations! Your manuscript is now being handed over to our production team.

Kind regards, 

on behalf of

Dr. Shahabedin Rahmatizadeh 

Academic Editor

PLOS ONE